# Truncation of NS1 Protein Enhances T Cell-Mediated Cross-Protection of a Live Attenuated Influenza Vaccine Virus Expressing Wild-Type Nucleoprotein

**DOI:** 10.3390/vaccines11030501

**Published:** 2023-02-21

**Authors:** Polina Prokopenko, Victoria Matyushenko, Alexandra Rak, Ekaterina Stepanova, Anna Chistyakova, Arina Goshina, Igor Kudryavtsev, Larisa Rudenko, Irina Isakova-Sivak

**Affiliations:** Institute of Experimental Medicine, 197022 Saint Petersburg, Russia

**Keywords:** influenza, live attenuated influenza vaccine, T-cell epitope, nucleoprotein, nonstructural protein 1, truncated NS1, cross-reactivity, memory T cells, lung tissue-resident T cells

## Abstract

Current seasonal influenza vaccines have suboptimal effectiveness, especially in seasons dominated by viruses that do not match the vaccine. Therefore, finding new approaches to improve the immunogenicity and efficacy of traditional influenza vaccines is of high priority for public health. Licensed live attenuated influenza vaccine (LAIV) is a promising platform for designing broadly protective vaccines due to its ability to induce cross-reactive T-cell immunity. In this study, we tested the hypothesis that truncation of the nonstructural protein 1 (NS1) and the replacement of the nucleoprotein (NP) of the A/Leningrad/17 master donor virus with a recent NP, i.e., switching to 5:3 genome composition, could improve the cross-protective potential of the LAIV virus. We generated a panel of LAIV candidates differing from the classical vaccine by the source of NP gene and/or by the length of NS1 protein. We showed that NS1-modified LAIV viruses had reduced viral replication in the respiratory tract of mice, indicating a more attenuated phenotype compared to the LAIVs with full-length NS1. Most importantly, the LAIV candidate with both NP and NS genes modified induced a robust systemic and lung-localized memory CD8 T-cell response targeting more recent viruses, and better protected immunized mice against lethal challenge with a heterosubtypic influenza virus than the control LAIV variant. Overall, these data indicate that the 5:3 LAIVs with truncated NS1 may be beneficial for protection against heterologous influenza viruses and warrant further preclinical and clinical development.

## 1. Introduction

Influenza viruses are highly contagious respiratory pathogens that pose a constant threat to the global community. Annual influenza epidemics cause 3 to 5 million cases of severe respiratory illness, up to 650,000 of which are fatal [1]. The most effective means of controlling influenza is vaccination, which is mainly focused on reducing morbidity and preventing severe cases of influenza and its complications. Currently there is a great variety of influenza vaccines, but their common drawback is their narrow specificity, resulting in the need for annual updating of the strain composition and not always satisfactory immunogenicity and, therefore, effectiveness [2,3]. Therefore, finding new approaches to improve the immunogenicity and efficacy of seasonal and pandemic influenza vaccines is of high priority for public health.

Among the great variety of influenza vaccines used in public health practice and at different stages of development, the live attenuated influenza vaccine (LAIV) plays a special role. The LAIV was first developed in the Russian Federation and registered in 1987, and then a similar LAIV was licensed in the US in 2003 [4,5,6]. The great advantage of live attenuated influenza vaccine in comparison with inactivated influenza vaccine (IIV) is the painless, needle-free intranasal route of administration, which enables the formation of not only a systemic humoral and T-cell immune response, but also a potent mucosal immunity at the port of virus entry [7,8,9,10]. In contrast to IIV, live attenuated vaccines induce herd immunity, which is especially important in pediatric organized groups, and can also protect against drifted variants of the influenza virus [11].

The broad spectrum of the adaptive immune responses generated by live attenuated influenza vaccines makes these viruses a promising platform for the development of a highly effective vaccine capable of protecting humans against a wide range of influenza viruses. The nonstructural influenza virus protein NS1 is a multifunctional protein involved in various stages of virus–cell interaction: it is an antagonist of the antiviral cell response and a regulator of viral and cellular gene expression [12,13]. In particular, the NS1 protein of the influenza virus performs the activity of an interferon antagonist, and thus contributes to the development of a productive infection, impairing one of the most important arms of antiviral immunity [14]. Furthermore, the C-terminus of the NS1 protein has been attributed to a reduction of dendritic cell activation and, consequently, a decrease in stimulation of naïve T cells [15]. Significant evidence has been accumulated on the ability of influenza viruses to express the truncated NS1 protein to induce a high level of adaptive immune response, while also making the virus more attenuated [15,16,17,18,19,20,21]. However, the vast majority of studies have used the model laboratory strain A/Puerto Rico/8/34 (H1N1) or were generated on the basis of wild-type influenza virus, which has the significant disadvantage of a potential reversion to a virulent phenotype in case of possible reassortment with other circulating viruses. In this regard, the use of a licensed LAIV backbone for generating NS1-truncated variants is a promising strategy for developing safe influenza vaccine with enhanced cross-protective properties.

Another way to improve the cross-protective potential of LAIVs is to optimize the T-cell repertoire that is induced upon vaccination. Traditional LAIV viruses possess 6:2 genome composition, meaning that 6 genes originate from the attenuated backbone virus, while 2 genes coding for hemagglutinin (HA) and neuraminidase (NA) come from circulating influenza virus. It is known that nucleoprotein (NP) is the main target of T-cell immunity to influenza virus [22], and, considering that the LAIV’s NP gene originates from the cold-adapted master donor virus A/Leningrad/17, isolated in 1957, it is likely that some LAIV-induced T cells may no longer recognize epitopes of currently circulating influenza viruses. Indeed, mutations in CD8+ T-cell epitopes of the NP have been demonstrated to be associated with escape from virus-specific cytotoxic T lymphocytes (CTLs) [23,24]. Using PBMCs isolated from HLA-typed blood donors, we demonstrated that LAIVs expressing the nucleoprotein of a recent isolate could stimulate human influenza CD8+ T cells more relevant to current infections than classical LAIVs with 6:2 genome [25]. Experiments in mice confirmed that wild-type NP gives the vaccine strain an advantage in inducing a more relevant repertoire of T cells that recognize currently circulating viruses, whereas classical strains can produce excessive levels of T cells to outdated epitopes [26,27].

In the current study, we combined these two approaches for improving LAIV’s cross-protective potential and generated LAIV reassortant viruses with 6:2 and 5:3 genome formulations expressing full-length or truncated NS1 proteins, and performed immunogenicity and cross-protection studies in C57BL/6J mice.

## 2. Materials and Methods

### 2.1. Cells, Viruses and Peptides

Madin Darby canine kidney (MDCK, ATCC CCL-34) and Vero (ATCC CCL-81) cells were maintained in Dulbecco’s modified Eagle’s medium (DMEM) supplemented with 10% fetal bovine serum (FBS) and antibiotic-antimycotic solution (AA) (all from Capricorn, Düsseldorf, Germany).

The H7N9 LAIV viruses with 6:2 and 5:3 genome compositions expressing full-length NS1 proteins were previously generated by the means of reverse genetics on the A/Leningrad/17 backbone and HA and NA genes of A/Anhui/1/2013 (H7N9) virus [27]. To generate recombinant influenza viruses expressing truncated NS1 protein, three consecutive Stop codons were introduced after the 126th residue of the Len/17 NS1 protein open reading frame: nucleotides 402-AAGAACATC-410 were changed to 402-TAATGATAA-410. Site-directed mutagenesis was performed on the Len/17 NS gene using the Q5^®^ Site-Directed Mutagenesis Kit (New England Biolabs, Ipswich, MA, USA). Primers for the mutagenesis procedure were ordered from Evrogen Ltd. (Moscow, Russia). Infectious LAIV viruses with desired genome compositions were rescued using Len/17-based reverse genetics as previously described [28], with the exception that electroporation with the Neon Transfection system (Thermo Scientific, Waltham, MA, USA) was carried out to transfect Vero cells with required plasmid sets. The viruses were amplified in eggs and stored at −70 °C in aliquots. Overall, four LAIV candidates were evaluated in this study (Figure 1). For assessing mouse antibody responses, H7N9 LAIV 6:2 and H7N9 LAIV 5:3 viruses were concentrated using a standard procedure of ultracentrifugation of harvested allantoic fluid on a 60%/30% sucrose gradient.

A 7:1 PR8-based reassortant influenza virus bearing NP gene of the A/Anhui/1/2013 (H7N9) virus was additionally generated for this study to assess the protective potential of the NP-specific T cells. The PR8/H7N9 7:1 virus was rescued in Vero cells and amplified in eggs as described above. The identity of the genomes of all rescued influenza viruses was confirmed by full-genome Sanger sequencing using BigDye Terminator v3.1 Cycle Sequencing Kit (Thermo Scientific, USA).

Peptides corresponding to H-2^b^-restricted NP_366_ epitopes of Len/17 (ASNENMDTM) and Anhui (ASNENMEAM) viruses were chemically synthesized by Almabion Ltd. (Voronezh, Russia), with a purity of over 95%, as shown by high-performance liquid chromatography (HPLC). Peptides were reconstituted in dimethyl sulfoxide and stored at –70 °C in single-use aliquots.

### 2.2. Recombinant Proteins

The recombinant NP and NS1 proteins of Len/17 virus were expressed in E. Coli strain BL21 (DE3) using a pETDuet-1 vector. Viral RNA was extracted by a Biolabmix RNA Isolation Kit (Biolabmix, Novosibirsk, Russia) and genes encoding full-length NP and NS1 proteins accompanied with polyhistidine tags were amplified with a Qiagen OneStep RT-PCR Kit (Qiagen, Germantown, MD, USA). The expression constructs and pETDuet-1 vector were then restricted by BamHI and NotI enzymes and ligated using a T4 DNA Ligase Kit (Evrogen, Moscow, Russia). The bacteria were heat-shock transformed by the obtained plasmids, followed by selection by ampicillin resistance, colony PCR and Sanger sequencing. The overnight cultures of producer strains in LB medium containing 100 µg/mL ampicillin were diluted 1:100, grown for 2 h at 37 °C, then twice diluted and induced by 0.5 mM isopropyl β-d-1-thiogalactopyranoside (IPTG) for 4 h at the same temperature. The cells were settled at 4600 rpm, resuspended in ice-cold tris-buffered saline containing 5 mM imidazole, and lyzed by sonication. The obtained lyzates were clarified at 12,000× *g* for 15 min and loaded into the column containing Ni-NTA His-Bind Resin (Novagen, Madison, WI, USA) equilibrated with resuspension buffer. NP and NS1 proteins were eluted with buffer containing 1 M imidazole, dialyzed against PBS and stored at −70 °C. Identification of the recombinant proteins was performed by recognition using commercial monoclonal antibodies NS1-23-1 (Santa Cruz Biotechnology, Heidelberg, Germany) or A/NP/6D11 (EPDP Ltd., Saint Petersburg, Russia).

### 2.3. In Vitro Studies of Recombinant LAIV Viruses

Infectious activity of LAIV viruses in eggs incubated at different temperatures was determined to assess temperature-sensitive (*ts*) and cold-adapted (*ca*) phenotypes of the recombinant viruses. Virus titer at 38 °C was compared to the titer at optimal temperature 33 °C for the *ts* phenotype, and 26 °C was compared with 33 °C for the *ca* phenotype. In addition, viral growth properties were assessed on MDCK and Vero cells using a 96-well plate format. Virus titers in eggs and MDCK cells were calculated by the Reed and Muench method and expressed in terms of log10 50% egg infective dose (EID_50_)/mL and log_10_TCID_50_/mL, respectively.

### 2.4. SDS-PAGE and Western Blot

Western blot analysis was performed to detect viral proteins in virus-infected MDCK cells. To prepare virus-infected MDCK cell lysates, 24-well plates were seeded with 2 × 10^5^ cells per well the day before virus inoculation. Cell monolayers were washed with PBS and inoculated with studied viruses at a multiplicity of infection 0.01 in DMEM (Capricorn, Germany) supplemented with 1 μg/mL TPCK-treated trypsin (Sigma, St. Louis, MO, USA). After adsorption at 33 °C for 1 h, the virus inocula were replaced with DMEM supplemented with TPCK trypsin, and the cells were incubated at 33 °C for 3 days. Then maintaining media were removed, the cells were washed with PBS and treated with ice-cold RIPA buffer (comprising 150 мМ NaCl, 50 мМ tris-HCl, 1 мМ EDTA, 1 мМ PMSF, 1% triton X-100, 0.1% SDS, pH 8.0) for 10 min on ice. The collected lysates were centrifuged at 12,000× *g* for 15 min at 3 °C and supernatants were stored at −20 °C until used for sodium dodecyl sulphate polyacrylamide gel electrophoresis (SDS-PAGE) analysis.

SDS-PAGE was carried out according to the previously described method [29]. A running 12.5% polyacrylamide gel prepared on 50 mM Tris-HCl buffer (pH 6.8), a stacking 5% polyacrylamide gel based on 0.375 mM Tris-HCl buffer (pH 8.8) and electrode buffer containing 25 mM Tris-HCl and 192 mM glycine (pH 8.3) were used. The samples were prepared by addition of a loading buffer containing 0.125 M Tris-HCl, 4% SDS, 20% glycerol, 0.02% bromophenol blue. To perform the analysis in reducing conditions, we added 2% β-mercaptoethanol (pH 6.8) to the samples and boiled for 5 min after mixing. Separation was performed at 100 V per gel for 20 min and then at 200 A until the end of electrophoresis. The gels then were rinsed with distilled water and stained with Coomassie G-250 solution for 30 min at room temperature by a standard protocol, or processed using the western blotting method [30]. In the latter case, nitrocellulose membranes with a pore diameter of 0.45 μm wetted with a buffer containing 25 mM Tris-HCl, 192 mM glycine, and 20% methanol (pH 8.3) were used for semi-dry protein transfer for 2 h at 100 mA. Then the membranes were blocked with 5% skim milk on PBS-T for 1 h at 37 °C and overnight-incubated with a primary mouse monoclonal anti-NS1 antibody (NS1-23-1, Santa Cruz Biotechnology, Germany) diluted 1:200 in blocking buffer, or anti-NP antibody A/NP/6D11 conjugated with horseradish peroxidase (HRP) (EPDP Ltd., Saint Petersburg, Russia) diluted 1:200 at 4 °C. The next day, the membrane treated with anti-NS1 antibody was washed three times with PBS-T and treated with anti-mouse secondary antibodies conjugated with horseradish peroxidase (1:3000 in PBS-T, Bio-Rad, Hercules, CA, USA) for 1 h at 37 °C. Then, both anti-NS1 and anti-NP antibody-treated membranes were rinsed with distilled water and stained using a 0.05% solution of 3-amino-9-ethylcarbazole (Sigma, USA) in 50 mM sodium acetate buffer, pH 5.5, containing 1% hydrogen peroxide.

### 2.5. Mouse Immunization, Sample Collection and Challenge

Eight-to-ten-week old female C57BL/6J mice were purchased from the Stolbovaya mice breeding nursery (Moscow region, Russia). Mice were housed at the Animal Facility of the Institute of Experimental Medicine (Saint Petersburg, Russia) with free access to water and food. All mouse studies were approved by the Local Ethics Committee of the Institute of Experimental Medicine (approval number 1/20 dated 27 February 2020).

Groups of 25 mice were immunized intranasally (i.n.) with one of the four recombinant LAIV viruses at a dose of 10^6^ EID_50_ in a volume of 50 µL. A booster dose of each vaccine was administered three weeks after the first dose using the same doses and procedures. A group of control mice received two i.n. doses of PBS. Nasal turbinates and lungs were collected from four mice in each LAIV group to determine the level of vaccine virus replication in the mouse respiratory tract. For this, tissue homogenates were prepared in 1 mL of sterile PBS containing AA, using a small bead mill TissueLyser LT (QIAGEN, Hilden, Germany). After low-speed centrifugation, supernatants were collected and tittered by limiting dilutions in eggs. Viral titers were calculated by the Reed and Muench method and expressed in log_10_EID_50_/mL. Six mice from each experimental group were sacrificed on day 7 after the second immunization, and lungs and spleens were collected for T-cell analyses (see below).

To assess antibody immune responses, blood samples were collected from six mice from each immunization group via retro-orbital sinus on day 21 after the second vaccine dose. To assess cell-mediated immune responses, six mice from each group were euthanized on day 7 after the booster immunization, and spleens and lung tissues were aseptically collected after lung perfusion with 10 mL of PBS through the right ventricle.

For the challenge experiments, immunized mice were randomly divided into two groups: (i) for challenge with PR8/H7N9 7:1 at the dose of 6 log_10_EID_50_, which corresponded to 3 LD_50_ (n = 5); and (ii) for challenge with PR8/H7N9 7:1 at the dose of 7 log_10_EID_50_, which corresponded to 30 LD_50_ (n = 10). Infected mice were monitored and weighed daily during the 14 days after challenge. Mice that lost 70% of the original body weight were humanely euthanized and considered as succumbed to infection.

### 2.6. Assessment of Antibody Immune Responses

Serum IgG antibody levels to the whole H7N9 6:2 and H7N9 5:3 LAIV viruses were detected using enzyme-linked immunosorbent assay (ELISA) as previously described [31]. For this, high-binding 96-well plates (Corning, Glendale, AZ, USA) were coated with 50 ng/well of sucrose gradient purified whole influenza viruses in a carbonate-bicarbonate buffer at 4 °C overnight. After blocking with 5% skimmed milk for 1 h at 37 °C and washing 3 times with PBS-T, two-fold serum dilutions starting from 1:200 were added and the plates were incubated for 1 h at 37 °C. At the next step, washed plates were treated with goat anti-mouse antibody conjugated with HRP (BioRad, Hercules, CA, USA). The plates were developed using 1-Step Ultra TMB-ELISA Substrate Solution (Thermo Fisher Scientific, Waltham, MA, USA). After 15 min incubation at room temperature, the reaction was stopped by adding 1M H_2_SO_4_. The absorbance was measured at 450 nm using an xMark Microplate Spectrophotometer (BioRad, USA). Antibody levels were expressed as the area under the curve (AUC) of OD_450_ values for serum 2-fold dilutions from 1:200 to 1:51,200, using the trapezoidal rule.

### 2.7. Assessment of T-Cell Immune Responses

Systemic and lung-localized CD8+ memory T-cell responses were assessed as previously described [32], with some modifications. Briefly, single splenocytes were isolated in CR-0 media (RPMI-1640 supplemented with AA solution, 25 mM HEPES (all from Capricorn, Germany) and 50 μM 2-mercaptoethanol (Sigma, USA)), using 70 µm cell strainers (BD Biosciences, Franklin Lakes, NJ, USA). Red blood cells were then lysed with 1× Red blood cell lysis buffer (Biolegend, San Diego, CA, USA). For intracellular cytokine staining (ICS) of splenocytes, 2 × 10^6^ cells were added to sterile U-bottom well microplates in 100 µL of CR-10 media (CR-0 media containing 10% FBS). Then, 100 µL of CR-10 containing the mixture of 1 μM NP_366_ peptide with GolgiPlug™ solution (BD Biosciences, USA) was added to each well, to a final dilution of GolgiPlug 1:1000. Phorbol myristate acetate (PMA) stimulation (Sigma, USA) was used as a positive control; non-stimulated controls and isotype controls were also prepared. The cells were incubated for 5 h at 37 °C, 5% CO_2_, followed by staining for 20 min at 4 °C in the dark with live/dead fixable stain (ZombieAqua, Invitrogen, Waltham, MA, USA) and a mixture of the following fluorescently labelled surface antibodies: CD4-PerCP/Cy5.5, CD8-APC/Cy7, CD44-PE, and CD62L-BV421 (all from BioLegend, USA). A Cytofix/Cytoperm kit (BD Biosciences, USA) was used for fixation/permeabilization, followed by cell staining with intracellular cytokine antibodies IFNγ-FITC, TNFα-APC, and IL2-PE/Cy7 (all from BioLegend) for 20 min at 4 °C in the dark. Samples were fixed with Cytolast buffer (Biolegend) and analyzed by Navios flow cytometer (Beckman Coulter, Brea, CA, USA).

For the detection of lung tissue resident memory T cells (T_RM_), the perfused lungs were cut into small pieces with sterile scissors and treated with a mixture of DNAse I and collagenase (both from Sigma) for 40 min at 37 °C. Then a single-cell suspension was prepared using 70 µm cell strainers, and the red blood cells were lysed as described above. Stimulation of the cells with NP peptides was performed using the same procedure as for splenocytes, although a different antibody cocktail was used for staining for surface markers and intracellular cytokine: the surface antibody cocktail comprised CD4-PerCP/Cy5.5, CD8-APC/Cy7, CD44-APC, CD62L-BV421, CD69-PE/Cy7, and CD103-FITC (all from BioLegend, USA), while only intracellular cytokine was stained with antibody IFNγ-PE/Dazzle.

The number of cytokine-positive cells in stimulated groups were counted, and the spontaneous cytokine secretion level of non-stimulated controls subtracted.

### 2.8. Statistical Analyses

Data were analyzed with the GraphPad Prism 6.0 software (GraphPad Software Inc.). The statistical significance of virological and immunological outcomes was determined by one-way ANOVA followed by Tukey’s multiple comparisons test. Differences in the survival rates after influenza virus challenge were analyzed by a log-rank Mantel–Cox test. *p* values less than 0.05 were considered statistically significant.

## 3. Results

### 3.1. Generation of Recombinant LAIV Viruses with Different Genome Compositions and Assessment of Their Replicative Properties

To test the hypothesis that truncation of the NS1 protein and the replacement of the Len/17 virus NP with a recent NP could improve the cross-protective potential of LAIV virus, we studied four LAIV candidates: one classical H7N9 LAIV strain, two candidates with one modification, and one LAIV virus with both modifications in combination (Figure 1). The A/Anhui/1/2013 (H7N9) strain was used here as a model virus, because H7N9 LAIV could actively replicate in the respiratory tract of mice, inducing high levels of antibodies and T cells, and because of significant differences in the sequence of the immunodominant CTL epitope NP_366_ between Anhui (ASNENMEAM) and Len/17 (ASNENMDTM) viruses, which makes it possible to detect subtle differences in epitope-specific T cell levels between the two antigens.

All modified LAIV candidates replicated well at optimal temperature in eggs and possessed temperature-sensitive and cold-adapted phenotypes similar to the classical H7N9 6:2 NS1_full_ vaccine virus, although some reduction of infectious activity was noted for the H7N9 6:2 NS1_126_ strain, suggesting that truncation of NS1 protein may affect viral infectivity (Table 1). Nevertheless, this effect was not seen for the pair of H7N9 5:3 NS1_full_ and H7N9 5:3 NS1_126_ LAIV candidates, indicating that the source of NP gene is important for the NS1-mediated impairment of viral growth (Table 1). Most importantly, all four viruses had similar infectivity for mammalian cell cultures, regardless of their interferon competence status (Table 1).

All LAIV viruses were unable to replicate in the lungs of mice, confirming their attenuated phenotype. Only viruses with the full-length NS1 gene replicated in the nasal turbinates of mice, whereas viruses expressing the truncated NS1 protein were not detected in the respiratory tracts of animals, indicating their more attenuated phenotype compared to the candidates with full NS1 (Table 1).

### 3.2. Western Blot Analysis of NS1 Protein Expression

Since NS1 protein is not usually found in viral particles, we analyzed NS1 protein expression in MDCK cells infected with the studied viruses. The cells were infected with each of the four viruses at MOI 0.01, and the productive infection was confirmed by the detection of equal levels of NP expression for each studied virus (Figure 2a). Interestingly, the NP bands, both for cell lysates and the recombinant protein, could be detected by the A/NP/6D11 universal antibody only under non-reducing conditions, suggesting that the monoclonal antibody binds to the polymeric forms of NP and fails to bind thermo-dissociated NP monomeric subunits—a phenomenon previously reported for similar monoclonal anti-NP antibody N5D3 [33].

Importantly, when analyzing MDCK cell lysates using the monoclonal anti-NS1 antibody, the bands with an apparent molecular mass of approximately 25 kDa, corresponding to the viral NS1 protein, were detected only in lysates of cells that were infected with viruses expressing the full-length form of NS1 (Figure 2b). These data suggest that either the anti-NS1 antibody was specific to the C-terminal part of NS1, or the recognition epitopes of this antibody disappeared after NS1 truncation due to some conformational changes of the remaining NS1_126_ protein. Nevertheless, these results demonstrated that both LAIVs with modified NS genes do not produce functional NS1 proteins in infected cells and should be lacking some immune modulating functions that their C-terminal domains are responsible for.

### 3.3. Assessment of Humoral Immune Responses to the Studied LAIV Viruses

To assess immunogenicity of the rescued recombinant LAIV viruses, C57BL/6J mice were i.n. immunized twice with 10^6^ EID_50_ of each virus, at a three-week interval. Serum samples were analyzed in ELISA using H7N9 6:2 and H7N9 5:3 whole virus antigens. Two-dose immunization induced high levels of serum IgG antibodies in all study groups (Figure 3). At the same time, a shortening of the NS1 protein in the H7N9 6:2 vaccine strain resulted in higher levels of the antibodies against H7N9 6:2 virus (Figure 3A), while the difference between the H7N9 6:2 NS1_full_ and H7N9 6:2 NS1_126_ was not significant when H7N9 5:3 LAIV virus was used as antigen (Figure 3B). At the same time, the immunogenicity of H7N9 5:3 NS1_126_ LAIV candidate was maintained at the level of H7N9 5:3 with full-length NS1, regardless of the whole virus antigen used (Figure 3). Most likely, in this case, an increased immune response with NS shortening was not observed due to the influence of an unrelated NP gene within the genome of the reassortant vaccine virus, and further studies with other influenza A virus subtypes are required to confirm this assumption.

### 3.4. Assessment of Systemic and Lung-Localized CD8 T-Cell Responses to the Studied LAIV Viruses

LAIV reassortant viruses induced robust cytotoxic T lymphocyte (CTL) responses, as was determined by stimulating mouse splenocytes collected on the day after the booster immunization with immunodominant NP_366_ peptide. Importantly, only LAIVs with a 6:2 genome composition could induce significant levels of CD8+ memory T cells specific to the Len/17 NP_366_ epitope (Figure 4A,B). Remarkably, these virus-specific T cells were polyfunctional and secreted all three cytokines. Interestingly, when the NP from Len/17 was replaced by the NP of Anhui (H7N9) virus (i.e., LAIV viruses with 5:3 genome composition), dramatically reduced levels of Len/17 peptide-specific cells were observed after immunization (Figure 4A,B).

Opposite results were obtained when splenocytes were stimulated with the NP_366_ peptide from the recent Anhui/13 influenza virus. Here, multicytokine-producing cytotoxic T cells were detected in the 5:3 LAIV groups, but not in the 6:2 groups. In addition, significantly higher levels were observed in the group where both the NP and NS genes were modified (Figure 4A,C). These data indicate that 5:3 LAIV with the NP gene from recent influenza virus stimulates more adequate cytotoxic T-cell immunity than does classic 6:2 LAIV. Both 6:2 LAIV strains induced a strong CTL response, but these T cells no longer recognized the circulating virus and simply overloaded the immune system of the immunized individuals.

Next, we measured the proportion of IFNγ-secreting cytotoxic T cells in the lungs of immunized mice. Here, recapitulating the results observed for systemic T-cell immunity, vaccine strains with the 6:2 genome formula induced predominantly T cells recognizing only the old NP_366_ peptide (Figure 5A,B), whereas LAIV strains with the 5:3 genome formula were able to induce a T-cell response specific to the more recent NP_366_ epitope (Figure 5A,C). Again, the truncation of NS1 protein in the H7N9 5:3 LAIV virus resulted in a significant increase in the magnitude of virus-specific effector memory CD8 T-cell responses. But more importantly, most of these virus-specific CD8 T cells also possess a resident memory phenotype: most IFNγ-specific CTLs express T_RM_ markers CD69 and CD103 (Figure 5).

Overall, truncation of the NS1 protein can significantly increase levels of humoral and T-cell immune response to live attenuated influenza vaccine, and the incorporation of a more recent NP into the LAIV genome can significantly redirect CTL immunity towards viral epitopes more relevant to current infections.

### 3.5. Protective Potential of the LAIV Candidates against a Heterosubtypic Lethal Virus Challenge

The LAIVs tested in this study are based on H7N9 avian influenza virus, which is classified as a BSL-3 pathogen. Therefore, challenge experiments with H7N9 wild-type viruses could not be performed in our study. Nevertheless, to assess the protective role of NP-specific CTL responses, a heterosubtypic influenza virus was rescued which bears NP of the Anhui/13 (H7N9) virus and seven remaining genes of A/PR8/34 (H1N1) virus (PR8 7:1 reassortant strain). When this virus was administered at a lower dose of 10^6^ EID_50_, all mice from all four LAIV groups were fully protected against weight loss and lethality, whereas four of five mock-immunized mice succumbed to infection (Figure 6A). Strikingly, when the challenge dose on this PR8 7:1 reassortant virus was increased 10 times, significant weight loss was observed in all experimental groups, and immunization with the classic H7N9 6:2 strain as well as H7N9 5:3 strains expressing the full-length NS1 protein did not significantly increase animal survival, reaching only 30% (Figure 6B). Although the surviving mice from the H7N9 5:3 LAIV group fully recovered and gained much weight, this vaccine could not be considered protective due to the high mortality rate (Figure 6B). Shortening the NS1 protein open reading frame in the 6:2 LAIV virus also did not result in a statistically significant increase in the survival rate of mice compared to the PBS control group. However, combining the two approaches, i.e., shortening the NS1 protein to 126 amino acids and incorporating a recent NP into the LAIV genome, significantly reduced the lethality of the challenge virus, and the survival rate increased to 70% (Figure 6B). These data indicate that the high levels of Anhui NP_366_ epitope-specific CTLs most likely contributed to the protection against a lethal virus expressing NP of the H7N9 virus.

## 4. Discussion

Traditional influenza vaccines mainly target the surface antigens of hemagglutinin and neuraminidase, and vaccine-induced neutralizing antibodies are typically produced to the immunodominant hypervariable regions of these glycoproteins [34]. Due to the constantly evolving nature of influenza viruses, they easily escape from the action of these antibodies, rendering vaccination campaigns ineffective and requiring an almost annual update in the seasonal influenza vaccine formulations [35,36,37]. Multiple strategies for improving the immunogenic and protective properties of the licensed influenza vaccines have been proposed, and some of the developments have already reached the stage of clinical evaluation [38,39,40]. Although the ultimate goal of a truly universal influenza vaccine capable of inducing long-lasting immunity against all circulating and emerging influenza viruses has not yet been achieved, in the meantime, more affordable strategies to improve existing influenza vaccines should be considered. In this regard, live attenuated influenza vaccines appear to be a very promising platform for designing broadly protective vaccines, due to their intrinsic property of inducing cross-reactive antibodies and T cells, as well as mucosal immunity [41,42].

It is widely recognized that T cells are major contributors to immunity against influenza and other respiratory pathogens, which help clear pathogens from the organism [43,44]. Influenza virus nucleoprotein is a major target antigen for influenza-specific cross-reactive cytotoxic T cells [22,45], which makes this protein a promising target to use when designing T cell-based influenza vaccines [46]. To date, multiple vaccine candidates targeting conserved T-cell epitopes of influenza virus NP have been developed using different vaccine platforms [47,48,49,50,51,52] and some were shown to be immunogenic in humans [53,54,55,56,57].

Earlier, we proposed a simple and straightforward approach to optimizing the repertoire of T cells produced upon LAIV uptake by simply incorporating an additional wild-type gene into genome of LAIV reassortant viruses, i.e., by switching from the 6:2 genomic formula to 5:3 LAIVs [58]. This idea is based on the fact the NP undergoes slow evolutionary changes, and recent influenza viruses differ significantly from the Len/17 master donor virus in the composition of NP epitopes. At the same time, the NP gene of the Len/17 strain does not contain mutations responsible for the attenuated phenotype of the reassortant LAIV viruses and can be substituted with the wild-type NP without safety concerns [28]. Side-by-side comparisons of 6:2 versus 5:3 LAIVs of different subtypes in animal models confirmed that the LAIVs carrying wild-type NP were safe, immunogenic and had improved cross-protective potential against recent influenza viruses [26,27,59,60]. In the present study, we received additional evidence that the source of the NP gene is critical for the establishment of a CD8 T-cell repertoire that can recognize epitopes present in the currently circulating viruses. Thus, LAIVs with the 6:2 genome composition induced a robust CTL response to the homologous NP_366_ epitope, whereas very low reactivity to the more recent NP_366_ epitope was observed, indicating that classical LAIVs induce excessive levels of Len/17 NP epitope-specific CTLs that no longer recognize evolutionarily diverse viruses and unnecessarily overload the immune system of vaccinees. Interestingly, although Len/17-specific T cells were unable to recognize NP_366_ peptide of the Anhui virus, the Anhui-specific CTLs had some degree of cross-reactivity against the Len/17 NP_366_ epitope. This difference may be due to the varying efficiency of direct peptide loading into MHC class I molecules for these two NP_366_ epitopes. Furthermore, mutations arising in immunogenic epitopes during virus evolution represent an important mechanism which various pathogens use to evade immunity [61,62,63]. In our study, the mutated epitope can no longer be recognized by the T-cell immunity raised to the ancestral strain, whereas CTLs developed with the new NP_366_ epitope can still recognize the old epitope to some extent.

Next, we pursued the idea of enhancing T-cell immunity by truncating the NS1 protein of the LAIV reassortant virus, based on the knowledge that influenza virus NS1 protein is a bifunctional viral immunosuppressor which can inhibit both innate and adaptive immune responses by preventing type I IFN production and by attenuating DC maturation with a subsequent decrease in T cell activation, respectively [64]. Thus, modification of NS1 genes in wild-type influenza viruses results in a robust local type I interferon response upon viral infection, which limits viral replication and ultimately leads to disease attenuation in different animal models [65]. Based on this idea, several vaccines with the NS1 gene deleted or truncated have been developed and have shown increased immunogenicity and cross-protective potential in preclinical and clinical trials [66,67,68,69,70,71,72]. However, a major disadvantage of the NS1-based LAIVs is that there is a risk of reverting to the wild-type phenotype if the vaccinated individual becomes infected with a naturally occurring influenza infection, and the full-length NS gene can be acquired by the LAIV virus through a reassortment event. In contrast, the attenuated phenotype of the Len/17-based LAIV is controlled by multiple mutations in the polymerase genes; therefore reversion to a virulent phenotype is very unlikely [28].

Importantly, in our study, we observed a clear enhancement of adaptive virus-specific CD8 T-cell responses after intranasal immunization with NS1-modified LAIV_S_, which was particularly notable in the detection of lung-localized CD8 memory T cells: both 6:2 and 5:3 NS1-modified candidates had significantly better responses to the corresponding NP epitope than their counterparts. We also showed that the majority of the NP_366_-specific Tem cells in the lungs possessed resident memory phenotype (CD69+CD103+), which are assumed to be at the first line of defense against invading respiratory pathogens [73]. Therefore, it is critical that the specificity of T_RM_ cells match the currently circulating viruses; otherwise these CTLs will not be able to recognize cells infected with the virus. The results of the present study support the idea of switching from a 6:2 to 5:3 LAIV genome formula to induce a CTL response more appropriate to the current infection. Furthermore, to increase the magnitude of the induced CTL response, truncation of the NS1 protein is a very promising strategy.

Finally, the LAIV viruses with modified NP and NS genes have a potential for further improvement of their cross-protective properties. For example, several studies demonstrated that the HA molecule of the vaccine reassortant virus can be modified by incorporating tandem repeats of the conserved ectodomain of the M2 protein (M2e) at the N-terminus. Such chimeric LAIV viruses were shown to have identical phenotypic properties to the classical LAIV strain, and also induced significant M2e-specific antibody levels that were associated with enhanced protection against various heterosubtypic influenza virus infections [31,74,75,76]. It is likely that modifying all three genes of LAIV virus would generate more broadly protective vaccine, while maintaining the required attenuated phenotype and high replicative properties for cost-effective vaccine manufacturing.

## 5. Conclusions

In this study, we generated a panel of H7N9 LAIV candidates differing from the classical LAIV reassortant virus by the source of the NP gene and/or by the length of the virus-expressed NS1 protein. Importantly, these vaccine candidates were generated using cold-adapted master donor virus, which is used for the routine development of licensed LAIVs in Russia and other countries. We showed that modification of the LAIV strains by truncating the NS1 protein open reading frame and by incorporating the NP gene from the recent influenza virus into the LAIV genome had no negative effect on the replicative properties of the virus in vitro. In addition, when the NS1 protein was truncated, a decrease in viral replication activity in the respiratory tract of mice was observed, indicating a more attenuated phenotype of these viruses compared to the LAIV strains with full-length NS1. This may indicate the increased safety of the NS1-modified vaccines and the possibility of their use in children younger than three years of age. Most importantly, the LAIV candidate with both NP and NS genes modified induced a robust systemic and lung-localized memory CD8 T-cell response targeting more recent viruses, and better protected immunized mice against lethal challenge, with a heterosubtypic influenza virus than with the control LAIV variant. Therefore, these data indicate that the 5:3 LAIVs with truncated NS1 may be beneficial for protection against heterologous influenza viruses. This proof-of-concept study was based on avian H7N9 influenza virus subtype; however, a similar effect is expected for currently circulating seasonal H1N1 and H3N2 viruses whose NP sequences also differ significantly from the Len/17 strain.

## Figures and Tables

**Figure 1 vaccines-11-00501-f001:**
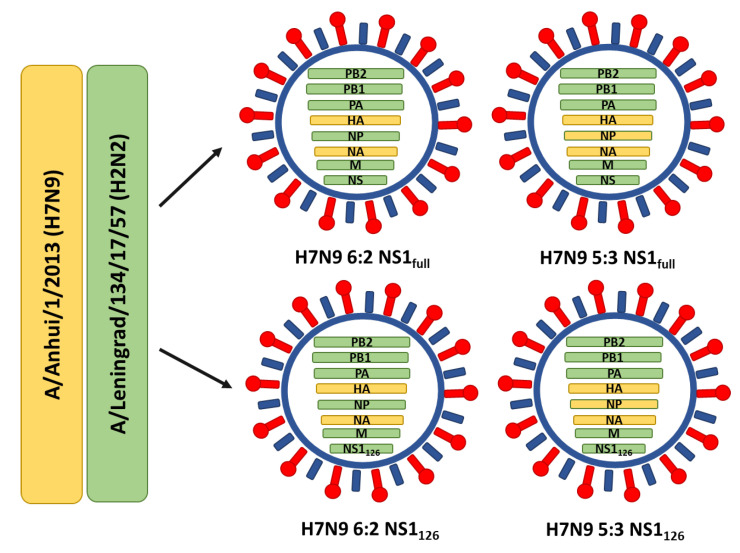
Overview of the LAIV reassortant viruses used in this study and their genome compositions.

**Figure 2 vaccines-11-00501-f002:**
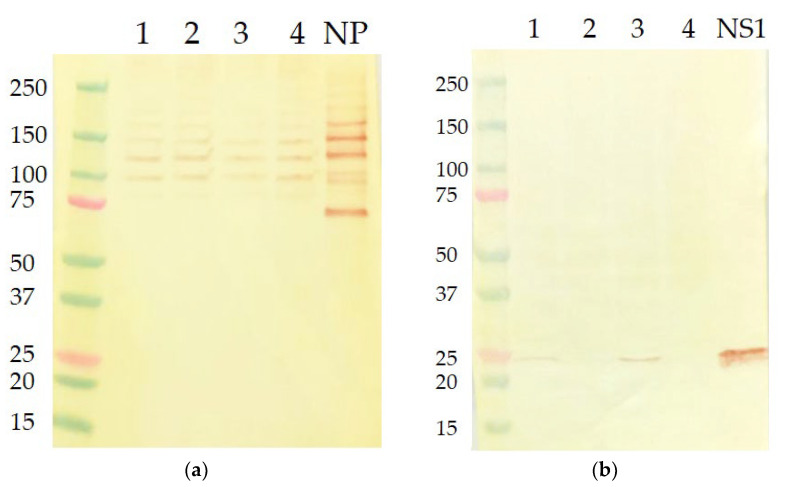
Analysis of NP expression in MDCK cells infected with recombinant LAIV viruses used in this study. (**a**) Western blot analysis using monoclonal anti-NP antibody after SDS-PAGE under non-reducing conditions; (**b**) Western blot analysis using monoclonal anti-NS1 antibody after SDS-PAGE under reducing conditions. 1—H7N9 6:2 NS1_full_; 2—H7N9 6:2 NS1_126_; 3—H7N9 5:3 NS1_full_; 4—H7N9 5:3 NS1_126_. NP: recombinant NP of Len/17 virus. NS1: recombinant NS1 protein of Len/17 virus.

**Figure 3 vaccines-11-00501-f003:**
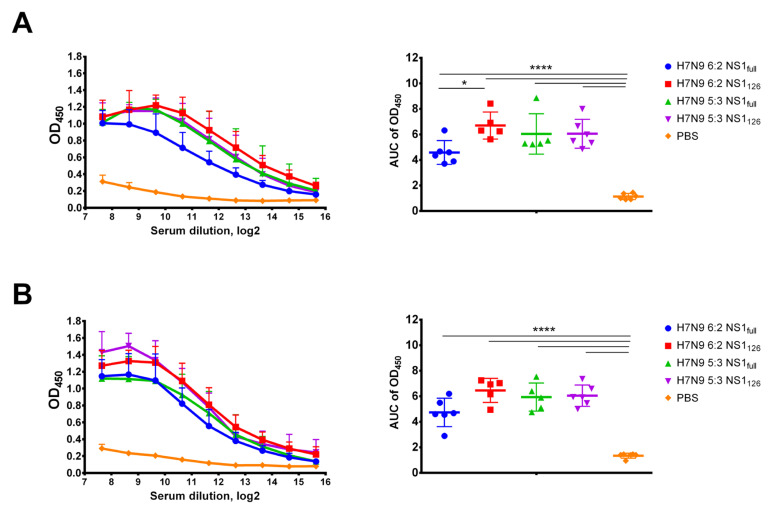
Serum IgG antibody immune responses to each LAIV candidate tested in this study. Mouse sera were collected three weeks after the second intranasal immunization. IgG antibody levels were assessed in ELISA with H7N9 6:2 LAIV whole virus (**A**) and H7N9 5:3 LAIV whole virus (**B**). Left panel shows mean OD_450_ values for each serum dilution. Right panel shows the area under the OD_450_ curve values for each individual animal. Data were analyzed by one-way ANOVA with Tukey’s post hoc multiple analyses test. * *p* < 0.05; **** *p* < 0.0001.

**Figure 4 vaccines-11-00501-f004:**
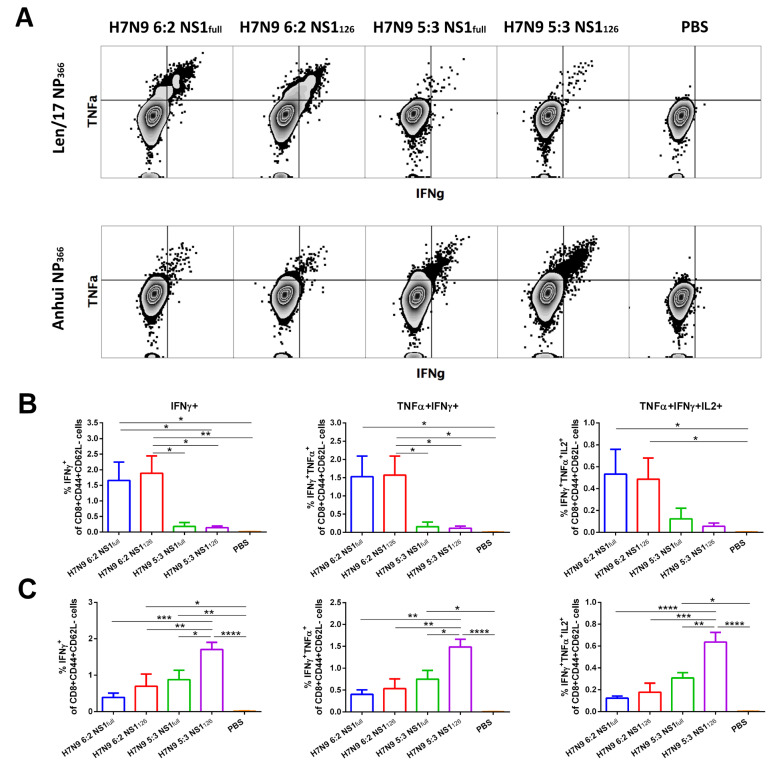
Systemic NP_366_-specific effector memory CD8+ T-cell responses, measured in mouse spleen on day 7 after the second immunization. (**A**) Representative flow cytometry plots for IFNγ and TNFα expression by Tem cells after stimulation with Len/17 or Anhui NP_366_ peptides. (**B**) Levels of CD8+ Tem cells expressing IFNγ only, IFNγ and TNFα or IFNγ, TNFα and IL-2 cytokines in response to stimulation with Len/17 NP_366_ peptide (ASNENMDTM). (**C**) Levels of CD8+ Tem cells expressing IFNγ, IFNγ and TNFα, or IFNγ, TNFα and IL-2 cytokines in response to stimulation with Anhui NP_366_ peptide (ASNENMEAM). Data were analyzed by one-way ANOVA with Tukey’s post hoc multiple analyses test. *—*p* < 0.05; **—*p* < 0.01; ***—*p* < 0.001; ****—*p* < 0.0001.

**Figure 5 vaccines-11-00501-f005:**
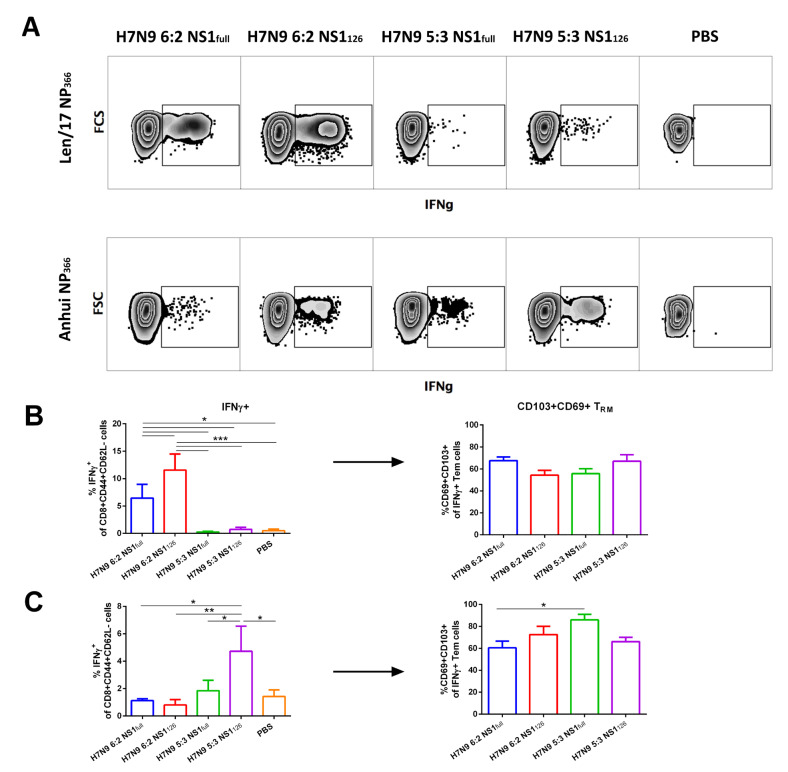
Lung-localized memory T-cell immune responses measured on day 7 after the second immunization. (**A**) Representative flow cytometry plots for IFNγ expression by Tem cells after stimulation with Len/17 or Anhui NP_366_ peptides. (**B**) Levels of CD8+ Tem cells expressing IFNγ cytokine in response to stimulation with Len/17 NP_366_ peptide (left graph) and the T_RM_ phenotypes of those IFNγ-positive effector memory T cells (right graph). (**C**) Levels of CD8+ Tem cells expressing IFNγ cytokine in response to stimulation with Anhui NP_366_ peptide (left graph) and the T_RM_ phenotypes of those IFNγ-positive effector memory T cells (right graph). Data were analyzed by one-way ANOVA with Tukey’s post hoc multiple analyses test. *—*p* < 0.05; **—*p* < 0.01; ***—*p* < 0.001.

**Figure 6 vaccines-11-00501-f006:**
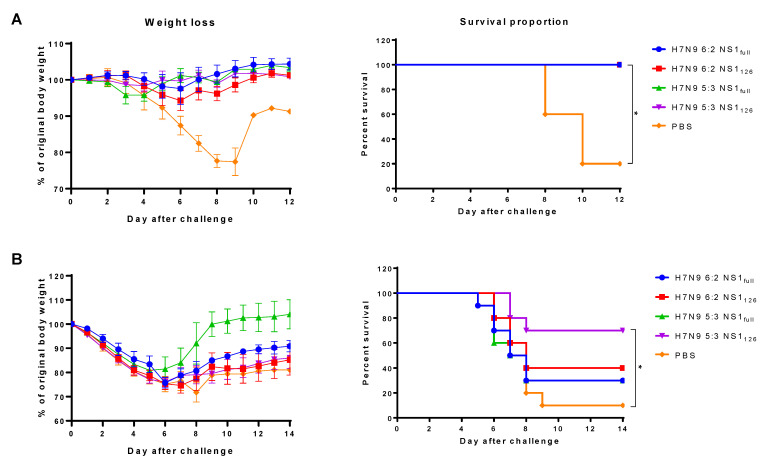
Protection of immunized mice against two different doses of heterosubtypic lethal virus. Mice were immunized with two doses of indicated H7N9 LAIV viruses or PBS and challenged with 10^6^ EID_50_ (**A**) or 10^7^ EID_50_ (**B**) of the PR8 7:1 (NP/H7N9) challenge virus. Weight loss (left panel) and survival rates (right panel) were monitored for two weeks post-challenge. Survival rates were compared by Mantel–Cox log-rank test. * *p* < 0.05.

**Table 1 vaccines-11-00501-t001:** Replicative properties of the recombinant LAIV viruses used in this study.

LAIV Candidate	Viral Titer in Eggs, log_10_EID_50_/mL			Viral Titer in Cells, log_10_TCID_50_/mL		Viral Titer in Mouse Tissues, log_10_EID_50_/mL	
	33 °C	38 °C	26 °C	MDCK	Vero	NT	Lung
H7N9 6:2 NS1_full_	9.4 ± 0.1	1.9 ± 0.3	5.9 ± 0.5	7.4 ± 1.6	6.1 ± 0.4	1.7 ± 0.5	<1.2
H7N9 6:2 NS1_126_	7.6 ± 0.1	<1.2	4.6 ± 0.7	6.0 ± 1.5	6.2 ± 0.7	<1.2	<1.2
H7N9 5:3 NS1_full_	8.7 ± 0.2	2.6 ± 1.2	6.3 ± 0.2	6.6 ± 0.7	6.4 ± 1.1	2.4 ± 1.4	<1.2
H7N9 5:3 NS1_126_	8.5 ± 0.3	2.9 ± 0.6	6.7 ± 0.7	6.0 ± 1.1	5.6 ± 0.3	<1.2	<1.2

NT: nasal turbinates.

## Data Availability

The data presented in this study are available on request from the corresponding author.

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
