# Peer review of "Truncation of NS1 Protein Enhances T Cell-Mediated Cross-Protection of a Live Attenuated Influenza Vaccine Virus Expressing Wild-Type Nucleoprotein"

_vaccines, 2023, doi:10.3390/vaccines11030501_

Round 1

Reviewer 1 Report

The manuscript described the modification of the classic LAIV backbone len/17, in which the NS1 is truncated to remove restriction on dendritic cell activation by the C-terminus of NS1. On Top of the truncated NS 1, the NP protein of the len/17 is replaced with that of A/Anhui/2013 (H7N9) aiming to preserve CTL responses to contemporary strains. As expected, truncated NS1 6:2 vaccine in mice induced higher antibody and T cells responses in the spleen and lungs than full NS1 6:2 to vaccine matched antigens, whereas truncated NS1 5:3 didn’t show advantage at inducing antibody responses, but enhanced CD8+ T cell responses to the matched NP peptide in the spleen and lungs. Again, as expected, 5:3 vaccine protected better against a hetertypic virus with a matched NP than 6:2 viruses.

Major comments:

-Figure 2, it looks like the first two lanes of 2a are pasted from another blot, which is not encouraged. Please re-run the gel. Also, it’s not clear what are the bands, which are very weak. Consider load more viruses and  please put in labels for each band.

-The authors should measure against 5:3 H7N9 NS full virus to look at if there is an increase of binding of NS126 sera.

-How conserved is the anhui NP366 epitope among relevant seasonal influenza virus strains?

-Figure 6, in the challenge study, the author use a hetertypic virus (PR8) with a matched NP to 5:3 viruses. However, the author aimed to induce a broader protection against the contemporary strains that are more relevant to human infection. Therefore, another virus with a unmatched NP which is from recent isolates should be tested

Author Response

The manuscript described the modification of the classic LAIV backbone len/17, in which the NS1 is truncated to remove restriction on dendritic cell activation by the C-terminus of NS1. On Top of the truncated NS 1, the NP protein of the len/17 is replaced with that of A/Anhui/2013 (H7N9) aiming to preserve CTL responses to contemporary strains. As expected, truncated NS1 6:2 vaccine in mice induced higher antibody and T cells responses in the spleen and lungs than full NS1 6:2 to vaccine matched antigens, whereas truncated NS1 5:3 didn’t show advantage at inducing antibody responses, but enhanced CD8+ T cell responses to the matched NP peptide in the spleen and lungs. Again, as expected, 5:3 vaccine protected better against a hetertypic virus with a matched NP than 6:2 viruses.

Authors’ response: we thank the reviewer for careful evaluation of our work and valuable suggestions.

Major comments:

-Figure 2, it looks like the first two lanes of 2a are pasted from another blot, which is not encouraged. Please re-run the gel. Also, it’s not clear what are the bands, which are very weak. Consider load more viruses and  please put in labels for each band.

Authors’ response: we thank the reviewer for this valuable critique. Since the main purpose of these WB experiments was to demonstrate the altered expression of NS1 protein by the NS1-modified LAIVs, we modified the setup of the experiment and did not include purified viruses in the analysis, since the NS1 protein is not incorporated into viral particles. From the other side, to prove that the cells infected with the studies viruses had productive infection, we performed WB analysis using anti-NP commercial monoclonal antibody. For this, SDS-PAGE under non-reducing conditions was used, since this mAb didn’t bind with NP monomers under reducing conditions. Furthermore, we were able to express control recombinant NP and NS1 proteins for this experiment. Overall, the figure 2 proves that the NS1 protein expression is compromised in cells infected with NS1-modified LAIVs. We added the corresponding information into results section, as well as the description of protein expression procedure (paragraph 2.2).

 -The authors should measure against 5:3 H7N9 NS full virus to look at if there is an increase of binding of NS126 sera.

Authors’ response: we thank the reviewer for this remark. We performed ELISA with this 5:3 antigen, but still didn’t find significant increase in the antibody level in the NS1-modified group. Probably, the source of NP protein antigen can affect the humoral immune responses generated by LAIV viruses. Further studies with H1N1 subtype LAIVs are ongoing to understand whether the immunogenicity of the NS1-modified vaccine viruses can be affected by the particular NP protein included into vaccine reassortant virus.

-How conserved is the anhui NP366 epitope among relevant seasonal influenza virus strains?

Authors’ response: we thank the reviewer for this question. The NP366 epitope differs by 2-3 residues with currently circulating H1 and H3 influenza A viruses. However, the aim of this study was not to develop a vaccine which could protect against recent seasonal influenza viruses, but to demonstrate that some differences in NP CTL epitopes between LAIV master donor virus and more recent viruses can have dramatic effect on T-cell immune responses and overall protective capacity against more recent viruses. This study involved H7N9 avian influenza virus isolated in 2013, and this model virus was selected for our proof-of-concept experiment because H7N LAIV could actively replicate in the respiratory tract of mice, inducing high levels of antibodies and T cells, and because of significant differences in the sequence of the immunodominant CTL epitope NP366 between Anhui (ASNENMEAM) and Len/17 (ASNENMDTM) viruses, which makes it possible to detect subtle differences in epitope-specific T cell levels between the two antigens (described in section 3.1). The main message of this study is that the NP protein of WT viruses should be incorporated into LAIV genomes, instead of Len/17 NP, and for each viral subtype (e.g. H1N1 and H3N2) their own NP should be included, to induce CTL immunity more relevant to current infections. We modified Conclusions section to emphasize the main message of the manuscript.

 -Figure 6, in the challenge study, the author use a hetertypic virus (PR8) with a matched NP to 5:3 viruses. However, the author aimed to induce a broader protection against the contemporary strains that are more relevant to human infection. Therefore, another virus with a unmatched NP which is from recent isolates should be tested

Authors’ response: we thank the reviewer for this note. The aim of this challenge experiment was to prove that the specificity of LAIV-induced T-cell responses may be critical for protection against recent influenza viruses. As noted above, the H7N9 viruses were used as a model strains, and the best choice would be to use the wild-type H7N9 virus to study protection in challenge experiment. However, this H7N9 strain is classified as BSL-3 virus, and we were unable to perform such challenge study. That’s why, the PR8-based challenge virus was specifically designed to make it as distant from the H7N9 virus as possible, only remaining the matched NP protein, to ensure that the NP-specific CTLs were responsible for protection. In our ongoing experiments we are testing H1N1 seasonal LAIV candidates with modified NP and NS1 genes, and the challenge viruses will correspond to recent human infections. We added the corresponding explanation to the section 3.5.

Reviewer 2 Report

In this very interesting manuscript, the authors designed a novel LAIV vaccine with truncated NS1 and updated NP. The idea is that the truncated NS1 will make viral infection less productive, while the updated NP sequence will induce an immune response that is more relevant to circulating strains. The authors measured the antibody as well as T cell response to test both ideas. Even though the data are not perfect, but the LAIV with both NS and NP modification had attenuated phenotype and induced lung-localized CD8 T-cell response targeting more relevant viral strains. Finally, the viral challenge experiment suggested that the newly designed LAIV can increase survival rates compared with control groups. Overall I think this manuscript should be accepted, but the following points should be addressed before publication:

1. For Figure 2, the authors should try using the anti-NS1 antibody for purified LAIV, and using the mouse sera for MDCK cell lysates. It’s possible that in Figure 2B the NS1 truncated groups have less virus in the lysate so the NS1 cannot be detected. 

2. In section 3.4, I don’t quite understand why Len/17 peptide almost cannot stimulate any T cell response for 5:3 LAIV groups but the Anhui/13 peptide can still stimulate some T cell response for 6:2 LAIV groups (Figure 4B vs 4C). Can this be explained by the sequences of NP? The authors should talk about this in the discussion session. 

Author Response

In this very interesting manuscript, the authors designed a novel LAIV vaccine with truncated NS1 and updated NP. The idea is that the truncated NS1 will make viral infection less productive, while the updated NP sequence will induce an immune response that is more relevant to circulating strains. The authors measured the antibody as well as T cell response to test both ideas. Even though the data are not perfect, but the LAIV with both NS and NP modification had attenuated phenotype and induced lung-localized CD8 T-cell response targeting more relevant viral strains. Finally, the viral challenge experiment suggested that the newly designed LAIV can increase survival rates compared with control groups. Overall I think this manuscript should be accepted, but the following points should be addressed before publication:

Authors’ response: we thank the reviewer for this positive feedback.

  1. For Figure 2, the authors should try using the anti-NS1 antibody for purified LAIV, and using the mouse sera for MDCK cell lysates. It’s possible that in Figure 2B the NS1 truncated groups have less virus in the lysate so the NS1 cannot be detected.

Authors’ response: we thank the reviewer for this note. As stated in the response to the Reviewer #1, we modified the setup of this experiment and did not include purified viruses in the analysis, since the NS1 protein is not incorporated into viral particles. From the other side, to prove that the cells infected with the studies viruses had productive infection, we performed WB analysis using anti-NP commercial monoclonal antibody. For this, SDS-PAGE under non-reducing conditions was used, since this mAb didn’t bind with NP monomers under reducing conditions. Furthermore, we were able to express control recombinant NP and NS1 proteins for this experiment. Overall, the figure 2 proves that the NS1 protein expression is compromised in cells infected with NS1-modified LAIVs. We added the corresponding information into results section, as well as the description of protein expression procedure (paragraph 2.2).

  1. In section 3.4, I don’t quite understand why Len/17 peptide almost cannot stimulate any T cell response for 5:3 LAIV groups but the Anhui/13 peptide can still stimulate some T cell response for 6:2 LAIV groups (Figure 4B vs 4C). Can this be explained by the sequences of NP? The authors should talk about this in the discussion session.

Authors’ response: we thank the reviewer for this valuable comment. Indeed, the degree of cross-reactivity of the NP366 epitope differs significantly between these two peptide variants. This difference may be due to the varying efficiency of direct peptide loading into MHC class I molecules for these two NP366 epitopes. Furthermore, mutations arising in immunogenic epitopes during virus evolution represent an important mechanism which various pathogens use to evade immunity. Here, the mutated epitope can no longer be recognized by the T-cell immunity raised to the ancestral strain, while CTLs developed to the new NP366 epitope can still recognize the old epitope to some extent. We added the following sentence to the Discussion section:

“Interestingly, although Len/17-specific T cells were unable to recognize NP366 peptide of the Anhui virus, the Anhui-specific CTLs had some degree of cross-reactivity against the Len/17 NP366 epitope. This difference may be due to the varying efficiency of direct peptide loading into MHC class I molecules for these two NP366 epitopes. Furthermore, mutations arising in immunogenic epitopes during virus evolution represent an important mechanism which various pathogens use to evade immunity [61-63]. In our study, the mutated epitope can no longer be recognized by the T-cell immunity raised to the ancestral strain, while CTLs developed to the new NP366 epitope can still recognize the old epitope to some extent”

Reviewer 3 Report

Known in the field based on previous literatures:

1.    Influenza is an infection of the respiratory system mainly include nose, throat, and lungs. Influenza is commonly called flu.

2. The vaccine comes in inactive and weakened viral forms and depending on the type they can be injected into a muscle, sprayed into the nose, or injected into the middle layer of the skin.

3.    Flu vaccine effectiveness can vary from year to year and the vaccine type, and even by virus type and subtype.

 In this article authors assessed and performed the following findings:

I have gone through the article titled “Truncation of NS1 protein enhances T cell-mediated cross-protection of a live attenuated influenza vaccine virus expressing wild-type nucleoprotein”. Authors nicely mentioned regarding influenza vaccine effectiveness and their drawback. They tested a hypothesis that truncation of the NS1 protein or the replacement of the NP protein of master donor virus with a recent NP, could improve the cross-protective potential of the licensed live attenuated influenza vaccine (LAIV) virus. The core points performed by authors are-

1.  They generated the recombinant LAIV viruses and evaluated their replicative properties.

2.    Authors measured the serum IgG antibody generated by LAIV viruses and further assessed the CD8 T-cell responses by the recombinant LAIV viruses.

The facts and material presented are interesting and generally supportive of the conclusions drawn. There are, however, some issues that require the authors' attention. The following suggestions if incorporated could help in the better understanding of the significance of the work and implications.

 Minor Concerns:

1.  Authors should write full meaning of all abbreviation when using first time. Please write abbreviation of NS1 and NP in abstract.

2. Authors escaped the method used in assessment of antibody immune responses. Briefly mention the method used by authors in 2.6.

3. Although, there are higher immune response generated in H7N9 6:2 NS1126, explain the higher TNFα level in H7N9 6:2 NS1full in the figure 4 (A). Please mention the value.

4. Why there is more body weight gained by H7N9 5:3 NS1full mice as compared to rest in figure (6B), although there is not significant change?

5.    Authors should clearly mention about how this article different from rest? Does it embrace a specific gap in the field as compared to previous?

Author Response

Known in the field based on previous literatures:

  1. Influenza is an infection of the respiratory system mainly include nose, throat, and lungs. Influenza is commonly called flu.
  2. The vaccine comes in inactive and weakened viral forms and depending on the type they can be injected into a muscle, sprayed into the nose, or injected into the middle layer of the skin.
  3. Flu vaccine effectiveness can vary from year to year and the vaccine type, and even by virus type and subtype.

 In this article authors assessed and performed the following findings:

I have gone through the article titled “Truncation of NS1 protein enhances T cell-mediated cross-protection of a live attenuated influenza vaccine virus expressing wild-type nucleoprotein”. Authors nicely mentioned regarding influenza vaccine effectiveness and their drawback. They tested a hypothesis that truncation of the NS1 protein or the replacement of the NP protein of master donor virus with a recent NP, could improve the cross-protective potential of the licensed live attenuated influenza vaccine (LAIV) virus. The core points performed by authors are-

  1. They generated the recombinant LAIV viruses and evaluated their replicative properties.
  2. Authors measured the serum IgG antibody generated by LAIV viruses and further assessed the CD8 T-cell responses by the recombinant LAIV viruses.

The facts and material presented are interesting and generally supportive of the conclusions drawn. There are, however, some issues that require the authors' attention. The following suggestions if incorporated could help in the better understanding of the significance of the work and implications.

Authors’ response: we thank the reviewer for careful evaluation of our work and valuable comments and suggestions.

 Minor Concerns:

  1. Authors should write full meaning of all abbreviation when using first time. Please write abbreviation of NS1 and NP in abstract.

Authors’ response: we thank the reviewer for this note. The full meanings were added to the abstract.

  1. Authors escaped the method used in assessment of antibody immune responses. Briefly mention the method used by authors in 2.6.

Authors’ response: we added the detailed description of the method into section 2.6.

  1. Although, there are higher immune response generated in H7N9 6:2 NS1126, explain the higher TNFα level in H7N9 6:2 NS1full in the figure 4 (A). Please mention the value.

Authors’ response: the figure 4A demonstrates only representative plots for each group, where the data from 1 animal was included. The higher TNFα level in H7N9 6:2 NS1full group was seen for this animal, but statistical analysis of the TNFa-producing T cells using full dataset didn’t find significant differences for this group relative to the other test groups.

  1. Why there is more body weight gained by H7N9 5:3 NS1full mice as compared to rest in figure (6B), although there is not significant change?

Authors’ response: the seemingly higher level of the body weight for this group was because the three surviving mice did not lose much weight, whereas the non-protected mice succumbed to infection early after challenge. In general, this vaccine cannot be considered protective due to the high mortality rate. We added the following sentence to the section 3.5 to address this issue: “Although the surviving mice from the H7N9 5:3 LAIV group fully recovered and gained much weight, this vaccine could not be considered protective due to the high mortality rate (Figure 6B).”

  1. Authors should clearly mention about how this article different from rest? Does it embrace a specific gap in the field as compared to previous?

Authors’ response: we thank the reviewer for this question. Our study differs from other published works in that we combined two approaches that can lead to increased immunogenicity and cross-protection - truncating the NS1 protein to 126 residues and replacing the vaccine NP with the recent NP (i.e., changing the genome formula from 6:2 to 5:3). Moreover, previous studies assessing influenza viruses with truncated or deleted NS1 mostly utilize a model influenza virus backbone: A/Puerto Rico/8/34 (H1N1), which has never been licensed for use as LAIV. In our study, a cold-adapted backbone for licensed LAIVs was used: A/Leningrad/17. Work on the replacement Len/17 NP with the wild-type NP has been done previously in our lab, but the combination of these two approaches is being reported here for the first time.

We added the following sentence to the Conclusions section to emphasize the novelty of the research: “Importantly, these vaccine candidates were generated using cold-adapted master donor virus which is used for routine development of licensed LAIVs in Russia and other coun-tries.”

Round 2

Reviewer 1 Report

The legend of figure 3 is not consistent with the revised figure, please modify accordingly

Author Response

The legend of figure 3 is not consistent with the revised figure, please modify accordingly

Authors' response: we thank the reviewer for noting this mistake. The figure legend was corrected accordingly.